# Spectral-Domain Optical Coherence Tomography Assessment in Treatment-Naïve Patients with Clinically Isolated Syndrome and Different Multiple Sclerosis Types: Findings and Relationship with the Disability Status

**DOI:** 10.3390/jcm10132892

**Published:** 2021-06-29

**Authors:** Łukasz Rzepiński, Jan Kucharczuk, Zdzisław Maciejek, Andrzej Grzybowski, Vincenzo Parisi

**Affiliations:** 1Department of Neurology, 10th Military Research Hospital and Polyclinic, Powstańców Warszawy 5, 85-681 Bydgoszcz, Poland; z.maciejek@wp.pl; 2Neurology Department, Sanitas—Neurology Outpatient Clinic, Dworcowa 110, 85-010 Bydgoszcz, Poland; 3Department of Ophthalmology, 10th Military Research Hospital and Polyclinic, Powstańców Warszawy 5, 85-681 Bydgoszcz, Poland; jankucharczuk@wp.pl; 4Department of Ophthalmology, University of Warmia and Mazury, Żołnierska 18, 10-561 Olsztyn, Poland; ae.grzybowski@gmail.com; 5Institute for Research in Ophthalmology, Foundation for Ophthalmology Development, Mickiewicza 24/3B, 60-836 Poznan, Poland; 6IRCCS—Fondazione Bietti, Via Livenza 3, 00198 Rome, Italy; vmparisi@gmail.com

**Keywords:** spectral optical coherence tomography, retinal layer segmentation, multiple sclerosis, disease type

## Abstract

This study evaluates the peripapillary retinal nerve fiber layer (pRNFL) thickness and total macular volume (TMV) using spectral-domain optical coherence tomography in treatment naïve patients with the clinically isolated syndrome (CIS) and different multiple sclerosis (MS) types. A total of 126 patients (15 CIS, 65 relapsing-remitting MS, 14 secondary progressive MS, 11 primary progressive MS, 21 benign MS) with or without optic neuritis (ON) history and 63 healthy age-similar controls were assessed. Concerning controls’ eyes, pRNFL thickness was significantly reduced in CIS-ON eyes (*p* < 0.01), while both TMV and pRNFL thickness was decreased in all MS eyes regardless of ON history (*p* < 0.01). Significant differences in pRNFL thickness and TMV between MS variants were observed for non-ON eyes (*p* < 0.01), with the lowest values in benign and secondary progressive disease type, respectively. The pRNFL thickness was inversely correlated with Expanded Disability Status Scale (EDSS) score in non-ON subgroups (*p* < 0.01), whereas TMV was inversely correlated with EDSS score in both ON and non-ON subgroups (*p* < 0.01). Concluding, pRNFL thinning confirms optic nerve damage in CIS-ON eyes and appears to be disproportionately high with respect to the disability status of benign MS patients. The values of TMV and pRNFL in non-ON eyes significantly correspond to MS course heterogeneity and patients’ disability than in ON eyes.

## 1. Introduction

Multiple sclerosis (MS) is a chronic, immune-mediated disorder of the central nervous system (CNS) with underlying inflammatory and degenerative mechanisms. The first clinical presentation of the disease, mainly in the form of optic neuritis (ON), focal supratentorial syndrome, focal brainstem/cerebellar dysfunction, or partial myelopathy, is referred to as clinically isolated syndrome (CIS) [1,2,3].

The majority of MS patients (85–90%) have an initial relapsing-remitting disease course (relapsing-remitting MS - RRMS) characterized by relapses associated with the formation of new demyelinating lesions in the CNS, followed by a complete or partial recovery and stable neurological condition between attacks. More than half of untreated RRMS subjects develop secondary progressive MS (SPMS) occurring as gradual disability progression regardless of superimposed bouts. Approximately 10–15% of patients experience steadily worsening neurological deficits from the disease onset, classified as primary progressive MS (PPMS) [2,4,5,6]. In the natural course of the disease, a benign type is distinguished much less often. Diagnosis of benign MS (BNMS) is determined retrospectively based on Expanded Disability Status Scale (EDSS) score and disease duration. However, available criteria consider different cut-off points [7,8,9,10]. MS variants differ in pathogenesis, the severity of neurodegeneration processes, and, most importantly, the rate of disability progression [11,12,13]. The heterogeneity of MS is also reflected in paraclinical studies. Magnetic resonance imaging (MRI) remains the most valuable, enabling monitoring of both inflammatory areas and brain atrophy. However, MRI imaging faces some limitations in monitoring subclinical aspects of the disease progression, such as normal-appearing white matter [14].

In recent decades, there has been an increase in interest in spectral-domain optical coherence tomography (SD-OCT) as a complementary diagnostic method for assessing neurodegenerative process in MS [1]. SD-OCT is a reproducible, non-invasive, high-resolution imaging technique for quantitative assessment of retinal layers [1,15,16,17,18,19,20]. Pathological findings confirmed the atrophy of inner retinal layers in different MS variants at all stages and variable clinical severity of the disease [21]. SD-OCT enables the visualization of axonal loss by measuring the peripapillary retinal nerve fiber layer (pRNFL) thickness and neuronal damage by assessing the volume or thickness of the macular area. Thus, SD-OCT is recommended for the diagnosis, monitoring, and research of MS [15,22].

Currently, most patients receive disease-modifying therapies (DMT) that ameliorate MS course and modulate retinal morphological changes [23]. Unfortunately, some reports have not addressed this issue or have not disclosed treatment data [1,22]. Furthermore, studies assessing the differences in SD-OCT measurements between MS types have revealed divergent opinions regarding the degree of pRNFL and retinal ganglion cells atrophy in the eyes of RRMS, SPMS, and PPMS subjects and their distinctness in BNMS eyes [1,24,25]. These diverging opinions could be partly a consequence of the high heterogeneity of the disease duration and the degree of patients’ disability between the cohorts assessed. However, the use of DMT constitutes an equally important aspect. Importantly, BNMS patients do not need aggressive treatments, and most studies contain different proportions of subjects using DMT, making it difficult to distinguish the benign course from therapy response [26]. So far, SD-OCT has not been included in MS diagnostic criteria. The International Panel on Diagnosis of MS identified studies concerning SD-OCT’s usefulness in detecting demyelinating lesion in the optic nerve of CIS patients as a high priority [3]. Therefore, the pRNFL thickness and macular area assessment limited to treatment-naïve patients with CIS and different MS types seem to be particularly valuable.

This study aimed to assess pRNFL thickness and total macular volume (TMV) in CIS and MS patients with different clinical characteristics (RRMS, SPMS, PPMS, and BNMS) who had never used DMT and to determine whether the changes in pRNFL or TMV values should be related to the EDSS score.

## 2. Materials and Methods

### 2.1. Participants

The study group consisted of 126 patients (90 females), with a mean age of 37.8 ± 10.2 years, and 63 volunteers (31 females), with a mean age of 35.8 ± 12.8 years (healthy controls—HCs), evaluated at an ophthalmological outpatient clinic between 2011 and 2014. Of the patients, 15 (10 females) had a diagnosis of CIS, and 111 patients (80 females) were diagnosed with MS, both established according to the revised 2010 McDonald criteria [27]. Eligible patients have never used DMT before SD-OCT evaluation. Clinical data on the duration of the disease, MS variant, history of ON, and the type of treatment used were obtained from available medical records. The disability was assessed based on the EDSS on the day of the SD-OCT examination [8]. The clinical course of MS at the time of the study was classified as RRMS, SPMS, PPMS (according to the Lublin and Reingold classification), or BNMS [4]. BNMS was defined as an EDSS score ≤3 after 15 years from the disease onset [9]. Disease duration was calculated from the occurrence of the first symptoms. Regarding the history of ON, only patients who experienced a single ON episode with at least a 6-month interval from symptom onset were included in the study. These criteria were chosen since it is known that the retrograde degeneration following ON occurs over 6 months [10]. For all study participants, the exclusion criteria were the presence of media opacity, retinal and choroidal diseases, CNS disorders other than MS, glaucoma, severe nystagmus or impaired eye movement that prevent eye fixation, the use of DMT, and any ophthalmological pathology that could modify the measures with SD-OCT. The study was approved by the Bioethics Committee of the Ludwik Rydygier Collegium Medicum (KB 532/2010). All patients gave their informed consent to storing their data in the database.

### 2.2. Spectral-Domain Optical Coherence Tomography (SD-OCT) Assessment

Retinal imaging was performed using the SD-OCT Copernicus HR device (OPTOPOL Technology, Zawiercie, Poland) with software version 4.3 and 3D measurement model used for calculations. SD-OCT was performed in each patient’s eye separately by the same examiner (K.J.). The quality of the scans was checked based on the OSCAR-IB [(O) obvious problems, (S) poor signal strength, (C) centration of scan, (A) algorithm failure, (R) retinal pathology other than MS related, (I) illumination and (B) beam placement] criteria, and the Advised Protocol for Optical Coherence Tomography Study Terminology and Elements (APOSTEL) recommendations were applied for data reporting [28,29]. Scanning of the macula and optic disc was performed in 6 × 6 mm quadrants with 512 × 128 scans. The pRNFL thickness was assessed using a ring scan with an internal diameter of 2.4 mm and a width of 0.4 mm. The middle of the ring was the center of the optic disc automatically determined by the software. TMV (mm^3^) was calculated by summarizing all the volumes obtained in the subfields using the inner, intermediate, and outer rings (with a diameter of 1 mm, 2.22 mm, and 3.45 mm, respectively). The areas of the retina where the pRNFL thickness and TMV were explored in the SD-OCT are shown in Figure 1.

### 2.3. Statistical Analysis

The Shapiro–Wilk test was used to check the compliance of the variable distribution with the normal distribution. In samples with a distribution close to normal, the results were presented as arithmetic means with standard deviation (SD), the student’s *t*-test was used for independent and dependent variables, and 1-way analysis of variance (ANOVA) was used to compare the means. Levene’s test assessed the homogeneity of variance. The Bonferroni test was used as a post-hoc test. When the distribution was significantly different from the normal distribution, the median, minimum, and maximum were calculated, and the significance of differences between the groups was checked using the non-parametric Mann–Whitney U test. Due to significant differences in the number of participants when comparing SD-OCT values between CIS and MS variants, the nonparametric ANOVA Kruskal–Wallis test was used. The relationship between SD-OCT parameters with the disease type and duration was evaluated using the Pearson correlation coefficient (r). Correlations of SD-OCT measurements with EDSS scores were calculated using ON history and adjusted for the disease duration. Both eyes of the patients and controls were analyzed. When verifying the statistical significance of differences between selected study groups, mixed-effects generalized linear models were fitted with an intrasubject correlation term. Correlation coefficients were estimated using a mixed-effects linear regression model with an intrasubject correlation term. In all tests, a significant conservative *p*-value of 0.01 was considered statistically significant. The *p*-value of 0.01 was chosen to represent a more significant impact than the value of 0.05 [30].

## 3. Results

### 3.1. Demographic and Clinical Features

The clinical and demographic data of the study population (CIS, RRMS, SPMS, PPMS, and BNMS groups) are presented in Table 1. Compared to other analyzed groups, BNMS patients were significantly older, had the longest disease duration, and had the longest time from the ON episode. The highest EDSS score was found in the PPMS group, and the highest percentage of participants with a history of ON was recorded in the SPMS group.

### 3.2. SD-OCT Peripapillary Retinal Nerve Fiber Layer Data

The results of SD-OCT pRNFL thickness measurements in the CIS, RRMS, SPMS, PPMS, and BNMS groups are presented in Table 2.

When the pRNFL thickness was considered based on the total, the values in the CIS group were not significantly (*p* > 0.01) reduced when compared to HCs group. The pRNFL thickness values detected in the RRMS, SPMS, and BNMS groups were significantly (*p* < 0.01) reduced with respect to those of HCs group.

When the pRNFL thickness was considered based on the history or absence of ON, the CIS subgroup with ON and the RRMS, SPMS, PPMS, and BNMS subgroups with and without ON showed a significant (*p* < 0.01) reduction of pRNFL thickness values concerning those of HCs group. In the RRMS, PPMS and BNMS subgroups with ON, a significant (*p* < 0.01) reduction of pRNFL thickness values regarding the same subgroups without ON were found. In CIS and SPMS, not significant (*p* > 0.01) differences between the non-ON (NON) and ON subgroups were found.

The pRNFL thickness in th eSPMS-NON and BNMS-NON subgroups was significantly reduced compared to CIS-NON subgroup (*p* = 0.0092 and *p* = 0.0002, respectively). There were no significant differences in pRNFL thickness between the CIS-NON, RRMS-NON, and PPMS-NON subgroups and between the CIS-ON and MS-ON subgroups regardless of the disease type (*p* > 0.01).

Statistically significant differences in pRNFL thickness were found between MS types only for NON subgroups (*p* < 0.01). The lowest pRNFL thickness was found in the BNMS-NON subgroup, and pRNFL atrophy was the least pronounced in the RRMS-NON subgroup (Figure 2).

### 3.3. SD-OCT Total Macular Volume Data

The SD-OCT TMV measurements in the CIS, RRMS, SPMS, PPMS and BNMS groups are presented in Table 2.

In the CIS group, when the TMV was considered based on total eyes, the values were not statistically (*p* > 0.01) different when compared to HCs ones. The TMV values detected in the RRMS, SPMS, and BNMS groups were significantly (*p* < 0.01) reduced concerning those of the HCs group.

When the TMV values were considered based on the history or absence of ON, both CIS subgroups with or without ON showed not significant (*p* > 0.01) differences compared to the HCs group. The RRMS, SPMS, and BNMS subgroups with and without ON showed a significant (*p* < 0.01) reduction of TMV values compared to the HCs group. In the BNMS, PPMS, and SPMS subgroups with ON, a significant (*p* < 0.01) reduction of TMV values concerning the same subgroups without ON was found. In CIS eyes, not significant (*p* > 0.01) differences between the subgroups with or without ON were found.

The TMV values found in the RRMS-NON, SPMS-NON, and BNMS-NON subgroups were significantly reduced compared to the CIS-NON subgroup (*p* = 0.0013, *p* = 0.0001 and *p* = 0.005, respectively). There were no significant differences in TMV values between the CIS-NON and PPMS-NON subgroups and between the CIS-ON and MS-ON subgroups regardless of the disease type (*p* > 0.01). Statistically significant differences in TMV were found between individual MS types only for NON subgroups (*p* < 0.01). Among all MS-NON eyes, the lowest TMV values were found in the SPMS subgroup, and the highest were found in the in PPMS subgroup (Figure 3).

### 3.4. Relationship between SD-OCT Data and Clinical Outcomes

There was no significant correlation of HCs age with pRNFL thickness and TMV (r = −0.18, *p* = 0.0464 and r = −0.20, *p* = 0.0293, respectively). Therefore, in CIS and MS patients, correlations of pRNFL thickness and TMV were calculated with the disease duration, taking into account the patients’ age as a control variable. For the CIS and RRMS, SPMS, PPMS, and BNMS groups, the relationships between SOCT measurements with the duration of disease and the EDSS score are presented in Table 3. Disease duration was correlated with pRNFL only in the PPMS group, whereas it was correlated with TMV in the PPMS and BNMS groups (Figure 4). Furthermore, disease duration was correlated with pRNFL thickness in the ON and NON subgroups (r = −0.36, *p* < 0.0001 and r = −0.34, *p* = 0.006, respectively), as well as with TMV in the ON and NON subgroups (r = −0.38, *p* < 0.0001 and r = −0.34, *p* = 0.006, respectively). After adjusting for disease duration, the EDSS score was significantly correlated with pRNFL values only in the NON subgroups. On the contrary, the EDSS score adjusted for disease duration was significantly correlated with TMV in the ON and NON subgroups (Figure 5). There were relationships between pRNFL and TMV in MS eyes regardless of the disease type. The strength of these associations was greatest in the BNMS and SPMS groups (r = 0.62, *p* < 0.0001 for RRMS; r = 0.85, *p* < 0.0001 for SPMS; r = 0.71, *p* < 0.0001 for BNMS; and r = 0.62, *p* = 0.0019 for PPMS). There was no significant correlation between pRNFL thickness and TMV in the CIS roup (r = 0.37, *p* = 0.0447) (Figure 6). The TMV values were correlated with pRNFL thickness in the ON and NON subgroups (r = 0.60, *p* < 0.0001 and r = 0.62, *p* < 0.0001, respectively).

## 4. Discussion

SD-OCT measurements are widely accepted biomarkers of pRNFL morphological involvement as well as neuronal and axonal involvement in MS [1]. Considering the confirmed protective effect of DMT on SD-OCT parameters, their assessment of the disease’s natural history is critical [31,32]. Our work aimed to assess the pRNFL thickness and TMV in treatment-naïve CIS and MS patients with different clinical characteristics and establish the relationship of these parameters with the EDSS score.

We found that, in CIS subgroups with or without ON, there were no significant (*p* > 0.01) differences of TMV values with respect to the HCs group, whereas a significant (*p* < 0.01) reduction of pRNFL thickness in the CIS-ON subgroup was observed. Among all analyzed MS subgroups with or without ON, significant (*p* < 0.01) reduction in pRNFL thickness and TMV values concerning the HCs were observed. The lowest values of pRNFL thickness and TMV were found in patients with BNMS and SPMS, respectively. The pRNFL thickness was significantly (*p* < 0.01) correlated with EDSS score only in the NON subgroups, whereas TMV was significantly (*p* < 0.01) correlated with EDSS score in both the ON and NON subgroups.

Optic nerve involvement can be a consequence of ON or subclinical neuroaxonal loss in the course of MS. Damage to the optic nerve myelin following ON causes secondary retrograde neurodegeneration of the axon and the retinal ganglion cell body. The extent of this damage, the degree of potential remyelination processes, and the mechanisms of subsequent axonal regeneration are important determinants of ON recovery and the death of the ganglion cell body. The remyelination capacity is most pronounced in the early demyelinating lesions and significantly decreases with chronic disease [33,34,35]. This mechanism may explain the thinning of pRNFL with typical TMV values in ON eyes of CIS patients who have a relatively short disease duration. On the other hand, pRNFL thinning in CIS-ON eyes compared to HCs eyes confirmed post-ON optic nerve damage, which may be useful in demonstrating dissemination in space (DIS) of demyelinating lesions.

The history of ON has the most significant influence on the reduction of pRNFL and TMV values. Thus, the highest percentage of BNMS and SPMS patients among those who experienced ON in our study group could significantly reduce pRNFL and TMV values. Interestingly, lower TMV values were found in SPMS patients characterized by more advanced disability. The obtained results are consistent with the study by Rothman et al. [36], who showed that lower baseline TMV significantly predicts higher disability. On the other hand, Thabit et al. [37] showed that pRNFL thickness is a sensitive correlate with the early disability in fully ambulatory MS patients, which undoubtedly includes subjects with BNMS.

The episode of ON may occur in 50% of MS subjects, and no differences in its course depending on the disease variant are observed [38,39]. Thus, no significant differences in pRNFL thickness and TMV between ON eyes of patients with various MS types can be found, as demonstrated in our study. On the other hand, frequent inner retinal layer involvement in PPMS patients despite the absence of acute ON episodes suggests that visual pathway injury in NON eyes is a part of the disease process [40]. Therefore, SD-OCT parameters in NON eyes seem to be more valuable for monitoring the heterogeneity of neuroaxonal loss and disability progression in MS, which was confirmed by our results. Our findings are related to the results of other authors with some remarks.

In a cross-sectional study, Eslami et al. [41] analyzed the differences between pRNFL and TMV in subjects with CIS, RRMS, and SPMS. The authors showed inverse correlations of both parameters with the EDSS score and an inverse correlation of pRNFL with the disease duration. However, patients with PPMS and BNMS were not included in this study, there were no HCs, and no information was provided on the type of DMT used. Furthermore, when assessing the correlation of SD-OCT parameters with clinical outcomes, history of ON was taken into account only for the disease duration. For this reason, a comprehensive comparison of these results with our findings is hindered. Interestingly, the authors found no significant differences in pRNFL thickness between NON and ON eyes of SPMS patients and in TMV between NON and ON eyes of CIS patients, which coincides with our results.

Oberwahrenbrock et al. [42] showed a significant pRNFL thickness reduction in SPMS patients compared to RRMS and a significant TMV decrease in SPMS and PPMS patients versus RRMS. Interestingly, the lowest TMV and pRNFL values in the eyes of SPMS patients, the differences in pRNFL measures between SPMS-NON and RRMS-NON eyes, as well as SPMS-ON and RRMS-ON eyes were similar to our findings (~10 μm versus ~5 μm in the study of Oberwahrenbrock et al., ~8 μm versus ~5 μm in our cohort). The pRNFL thickness difference of 5 μm between PPMS-NON and SPMS-NON eyes is in line with our data and previous reports [43,44]. Contrary to our results, the authors found an association of the disease duration with TMV only in RRMS eyes, correlation of pRNFL thickness with MS duration only in SPMS and PPMS eyes, and no significant correlation of EDSS score with TMV both for RRMS-ON and SPMS-ON eyes [42]. These differences can be explained by older age, approximately twice the duration of the analyzed MS variants, and a similar degree of disability in patients from the cohort of Oberwahrenbrock et al. compared to ours. However, it is not known whether the subjects described by Oberwahrenbrock et al. had a milder disease course or used DMT (information about the treatment was not disclosed).

Balk and colleagues [24] evaluated 230 patients with typical MS course (140 RRMS, 61 SPMS, 29 PPMS) and 63 HCs. Of typical MS participants, 59 patients were classified as BNMS. Data regarding the use of DMT in this cohort were not disclosed. The authors found significant differences in pRNFL and ganglion cell complex (GCC) thickness between MS types for NON eyes. Contrary to our findings, pRNFL and GCC thinning in BNMS-NON eyes were less pronounced than in typical MS-NON eyes. These differences can be explained by the significantly longer duration of MS variants and the different distribution of this time in the cohort described by Balk et al. (SPMS: 23 years, PPMS: 21.4 years, BNMS: 21.1 years, and RRMS: 19.1 years). We are also aware that, after 21 years from MS onset, some patients with BNMS from our study group could not fulfill the criterion of a benign course. However, caution should be taken when interpreting SOCT measurements in the absence of information on DMT.

In 11 BMNS, 25 RRMS, and 34 HCs subjects, Huang-Link et al. [10] showed significant pRNFL thinning both in RRMS and BNMS eyes and significant reduction of TMV only in RRMS eyes compared to control eyes. In this study, 26 MS patients (25 RRMS, 1 BNMS) received DMT, and BNMS was defined by an EDSS score of ≤2 after 10 years or ≤3 after 20 years from MS onset. Therefore, it cannot be excluded that DMT alleviated retinal atrophy and the BNMS cohort consisted of patients with low EDSS score. Additional confirmation of this assumption is the average EDSS score of one point and the lack of a significant difference in the disability level between RRMS and BNMS patients in this cohort. For these reasons, comparing these data to our findings has great limitations. In turn, a longitudinal study by Galetta and colleagues [25] evaluating 68 MS participants (13 with BNMS) showed similar degrees of pRNFL thinning in BNMS eyes compared to typical MS eyes. Furthermore, a history of ON correlated with pRNFL thickness at baseline and disease duration, but not EDSS. These results coincide with our observations, confirming no lower degree of retinal atrophy in BNMS versus conventional MS eyes and a better correlation of disability with SD-OCT parameters in NON than ON eyes. Unfortunately, the authors did not disclose the number of patients using DMT.

Bsteh et al. [45] assessed the predictive value of pRNFL loss for physical disability in a 3-year longitudinal study. Only NON eyes of 151 RRMS patients were evaluated, and the percentage of patients using DMT was 59.6% at baseline and 73% at the end of the follow-up period, respectively. The authors showed that pRNFL thinning was significantly negatively impacted by EDSS progression and disease duration, while DMT provided a positive influence. In terms of disability progression, these results are consistent with our findings and, at the same time, confirm the need to consider the effect of DMT on pRNFL atrophy.

Martinez-Lapiscina et al. [46] found a significant association of low pRNFL thickness at baseline with the increased risk of disability progression in 5 years of follow-up. The analysis considered only patients with no ON history, which may confirm the legitimacy of pRNFL monitoring in NON patients’ eyes demonstrated in our study. Contrary to our findings, the authors did not observe a meaningful association for macular volume calculated within a 6 mm ring area. TMV includes full retinal thickness, capturing both the ganglion cell layer and RNFL. Approximately 50% of retinal ganglion cells are concentrated within 4.5 mm of the fovea, and the macular parameter with the best diagnostic performance should be calculated for an annulus with an inner diameter of 3 mm and outer diameter of 4 mm [38,47]. In our study, TMV calculated for a 3.45 mm diameter ring area allowed an assessment of a greater proportion of ganglion cells than RNFL axons. Therefore, the TMV values obtained and their correlations with clinical results in our group may be different compared to studies assessing the macular volume in the larger diameter ring. Unfortunately, for treatment naïve CIS and MS patients from our cohort, evaluation of the macular area was possible by TMV and not by ganglion cell + inner plexiform layer (GCIP) thickness. It is one of the limitations of the presented study. A relatively low number of patients, especially with PPMS, could have affected the final differences in SD-OCT parameters between the analyzed subgroups, constituting a limitation of the presented study. Another limitation of the study is the lack of assessment of the relationship between MRI activity and the number of relapses with the obtained SD-OCT measurements.

Further limitations are the strongly heterogeneous study group, relatively short disease duration, and lack of prospective assessment of patients. Unfortunately, we did not explore why the patients we assessed were treatment-naïve, which is another limitation of the presented study. Since 2010, together with the progress of retinal segmentation methods, a rapid increase in DMT availability has been observed in Poland. Therefore, with the possibility of assessing GCIP thickness, the number of treatment-naïve MS patients has steadily decreased, which has also hindered the follow-up assessment [48]. Apart from evaluating patients without DMT and considering both CIS and the broad spectrum of MS variants, it is challenging to demonstrate innovation in other aspects of the presented study. Nevertheless, we believe that our findings can be a valuable contribution to assessing SOCT parameters in treatment-naïve CIS and MS patients, providing a certain reference point for subsequent studies in the era of widespread DMT availability.

## 5. Conclusions

The presented study confirms the varied involvement of inner retinal layers in treatment-naïve MS patients with different disease types only in the case of NON eyes. The pRNFL thinning appeared to be an indicator of optic nerve damage in CIS-ON eyes compared to HCs eyes, which may support the utility of including this parameter in demonstrating DIS of demyelinating lesions. Surprisingly, the results disproportionately emphasize the disability level of pRNFL thinning in the eyes of BNMS patients. Thus, the BNMS does not necessarily have to be benign with respect to the SD-OCT measurements. Importantly, our results refer to treatment-naïve patients, which emphasizes both the adequacy of BNMS definition and provides a better view of this MS variant in the context of other disease types. We speculate the need for further SD-OCT studies in BNMS patients to clarify or extend the definition of this MS variant. In the context of analyzing the disease course of treatment-naive CIS and MS patients, our findings can be useful for understanding the natural history of these disorders. Moreover, by demonstrating a better correlation of EDSS score with TMV and pRNFL thickness in NON eyes than in ON eyes, they can be helpful in everyday clinical practice.

## Figures and Tables

**Figure 1 jcm-10-02892-f001:**
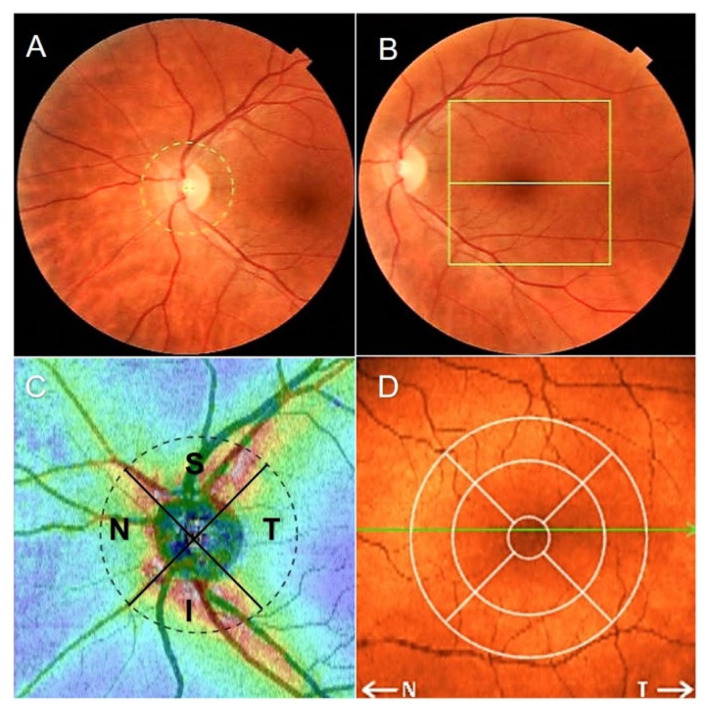
Spectral-domain optical coherence tomography (SD-OCT) report of the retinal thickness analysis. (**A**) Disc area scan, (**B**) macular area scan, and (**C**) peripapillary retinal nerve fiber layer quadrants (S: Superior; T: Temporalis; I: Inferior; N: Nasalis) on retina thickness map. (**D**) Concentric macular rings (1 mm, 2.22 mm, 3.45 mm) on retina map.

**Figure 2 jcm-10-02892-f002:**
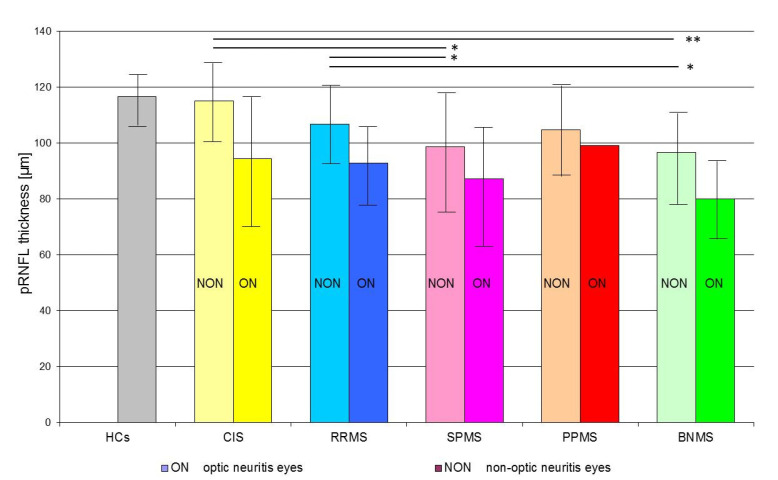
Mean values (bar graph), 1 standard deviation (vertical lines), and *p*-value (* < 0.01, ** < 0.001—horizontal lines) of peripapillary retinal nerve fiber layer (pRNFL) thickness in the investigated subgroups. Abbreviations: HCs—healthy age-similar controls, CIS—clinically isolated syndrome, RRMS—relapsing-remitting multiple sclerosis, SPMS—secondary progressive multiple sclerosis, PPMS—primary progressive multiple sclerosis, BNMS—benign multiple sclerosis.

**Figure 3 jcm-10-02892-f003:**
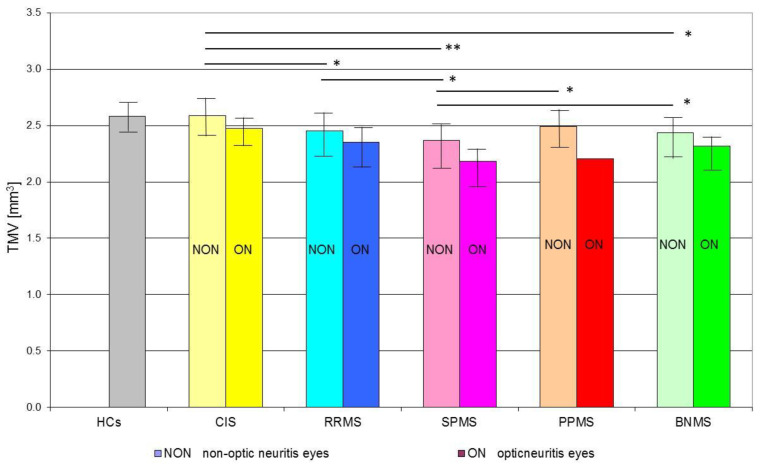
Mean values (bar graphs), 1 standard deviation (vertical lines), and *p*-value (* < 0.01, ** < 0.001—horizontal lines) of total macular volume (TMV) in the investigated subgroups. Abbreviations: HCs—healthy age-similar controls, CIS—clinically isolated syndrome, RRMS—relapsing-remitting multiple sclerosis, SPMS—secondary progressive multiple sclerosis, PPMS—primary progressive multiple sclerosis, BNMS—benign multiple sclerosis, ON—optic neuritis, NON—non-optic neuritis.

**Figure 4 jcm-10-02892-f004:**
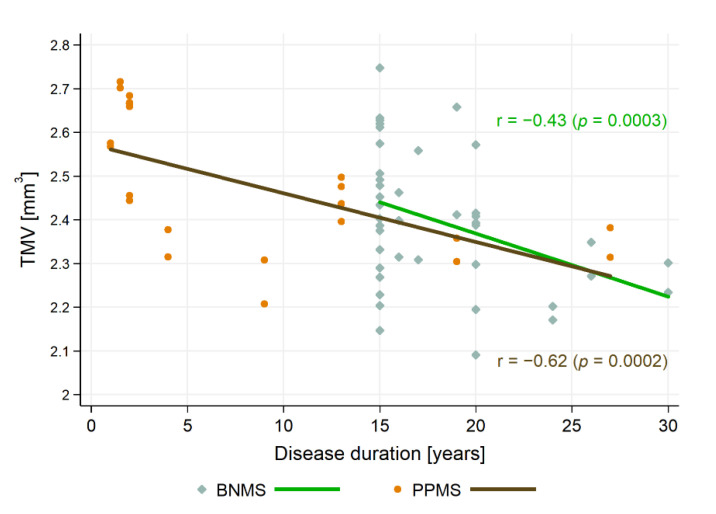
The relationships between total macular volume (TMV) and disease duration. Abbreviations: PPMS—primary progressive multiple sclerosis, BNMS—benign multiple sclerosis.

**Figure 5 jcm-10-02892-f005:**
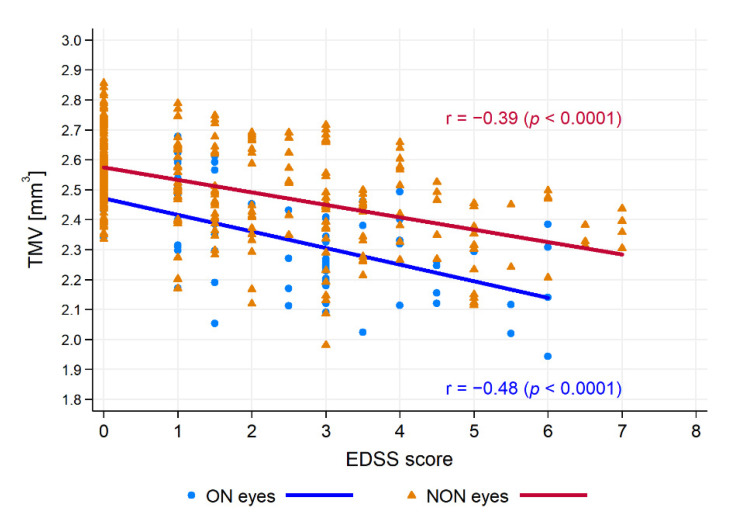
The relationships between total macular volume (TMV) and Expanded Disability Status Scale (EDSS) score. Abbreviations: ON—optic neuritis, NON—non-optic neuritis.

**Figure 6 jcm-10-02892-f006:**
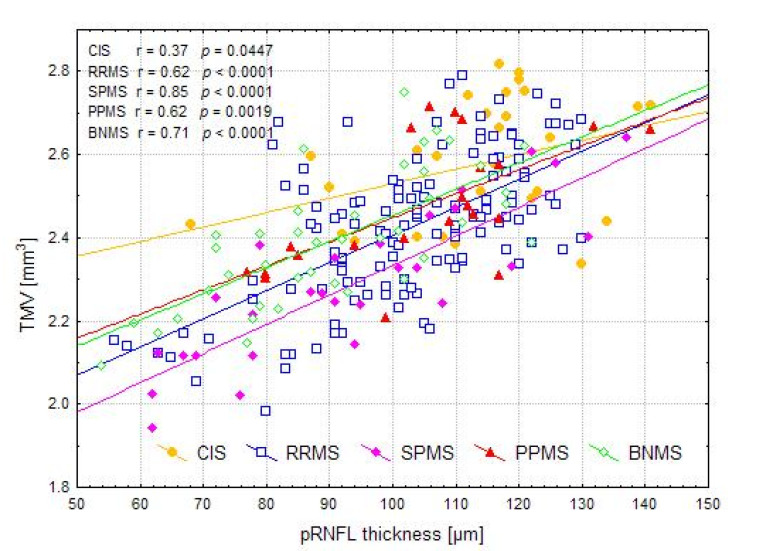
The relationships between peripapillary retinal nerve fiber layer (pRNFL) thickness and total macular volume (TMV) values in the investigated groups. Abbreviations: CIS—clinically isolated syndrome, RRMS—relapsing-remitting multiple sclerosis, SPMS—secondary progressive multiple sclerosis, PPMS—primary progressive multiple sclerosis, BNMS—benign multiple sclerosis.

**Table 1 jcm-10-02892-t001:** Demographic, clinical, and ophthalmic data of the studied patients.

	CIS	RRMS	SPMS	PPMS	BNMS	HCs	*p*-Value
No. of patients/eyes	15/30	65/130	14/28	11/22	21/42	63/126	
Sex (female/male)	10/5	45/20	12/2	8/3	15/6	31/32	0.0614
Mean age (range) [years]	35.3 ± 11.0 (20–55)	34.7 ± 8.8 (18–53)	39.1 ± 10.5 (24–59)	42.9 ± 10.7 (26–54)	45.5 ± 8.7 (31–66)	35.8 ± 12.8 (18–69)	0.001
Median disease duration(range) [years]	1 (0.3–4)	3 (0.5–23)	9.5 (4–22)	4 (1–27)	16 (15–30)		<0.0001
Median EDSS score (range)	0 (0–2)	2.0 (0–5)	4.3 (3–6.5)	5.0 (3–7)	2.0 (1–3)		<0.0001
ON/NON eyes (%ON)	3/27 (10%)	35/95 (26.93%)	12/16 (42.86%)	1/21 (4.55%)	14/28 (33.33%)		0.0146
Median time elapsed from ON(range) [months]	12 (6–24)	30 (12–90)	110 (60–150)	40	195 (180–240)		<0.0001

Abbreviations: CIS—clinically isolated syndrome, RRMS—relapsing-remitting multiple sclerosis, SPMS—secondary progressive multiple sclerosis, PPMS—primary progressive multiple sclerosis, BNMS—benign multiple sclerosis, EDSS—Expanded Disability Status Scale, ON—optic neuritis, NON—non-optic neuritis.

**Table 2 jcm-10-02892-t002:** Mean values and one standard deviation (±) of Spectral Optical Coherence Tomography peripapillary retinal nerve fiber layer (pRNFL) thickness and total macular volume (TMV) assessment. Results of statistical analysis between groups.

	Total Eyes	NON Eyes	ON Eyes		*p*-Value		
				HCs vs. Total	HCs vs. NON	HCs vs. ON	NON vs. ON
pRNFL (µm)							
HCs	116.5 ± 9.0						
CIS	113.0 ± 15.6	115.0 ± 13.7	94.3 ± 22.8	0.3341	0.6868	0.0001	0.0905
RRMS	102.9 ± 15.9	106.6 ± 14.9 #	92.8 ± 14.3 *	<0.0001	<0.0001	<0.0001	0.0001
SPMS	93.8 ± 21.7	98.7 ± 21.0 #	87.3 ± 21.8 *	<0.0001	0.0094	<0.0001	0.3135
PPMS	104.6 ± 16.8	105.2 ± 16.6 #	99.0 *	0.0193	0.0188	<0.0001	0.0018
BNMS	91.0 ± 17.4	96.6 ± 16.7 #	79.9 ± 13.3 *	<0.0001	<0.0001	<0.0001	<0.0001
TMV (mm^3^)							
HCs	2.580 ± 0.104						
CIS	2.575 ± 0.147	2.586 ± 0.149	2.476 ± 0.105	0.8943	0.8813	0.0911	0.0617
RRMS	2.426 ± 0.179	2.453 ± 0.165 §	2.352 ± 0.170 ℒ	<0.0001	<0.0001	<0.0001	0.0267
SPMS	2.289 ± 0.180	2.367 ± 0.163 §	2.185 ± 0.149 ℒ	<0.0001	0.0001	<0.0001	0.0090
PPMS	2.478 ± 0.154	2.490 ± 0.146 §	2.207 ℒ	0.0288	0.0396	<0.0001	<0.0001
BNMS	2.397 ± 0.159	2.437 ± 0.154 §	2.317 ± 0.129 ℒ	<0.0001	0.0001	<0.0001	0.0006

Abbreviations: HCs—Healthy Controls, CIS—clinically isolated syndrome, RRMS—relapsing-remitting multiple sclerosis, SPMS—secondary progressive multiple sclerosis, PPMS—primary progressive multiple sclerosis, BNMS—benign multiple sclerosis, ON—optic neuritis, NON—non-optic neuritis. Analysis of variance (ANOVA) between NON and ON MS subgroups: § *p* = 0.0002; # *p* = 0.0003, ℒ *p* = 0.0233, * *p* = 0.0555.

**Table 3 jcm-10-02892-t003:** Correlations between Spectral Optical Coherence Tomography parameters with selected clinical outcomes.

	pRNFL Thickness		TMV	
	Correlation Coefficient	*p*	Correlation Coefficient	*p*
CIS duration	r = −0.01	0.9490	r = −0.14	0.4570
BNMS duration	r = −0.32	0.0104	r = −0.43	0.0003
RRMS duration	r = −0.10	0.1217	r = −0.19	0.0290
SPMS duration	r = −0.16	0.2661	r = −0.19	0.2319
PPMS duration	r = −0.55	0.0016	r = −0.61	0.0002
EDSS score	R = −0.22	0.0004	R = −0.39	<0.0001
(a) ON eyes	R = −0.01	0.9364	R = −0.48	<0.0001
(b) NON eyes	R = −0.29	0.0003	R = −0.39	<0.0001

Abbreviations: pRNFL—peripapillary retinal nerve fiber layer, TMV—total macular volume, CIS—clinically isolated syndrome, RRMS—relapsing-remitting multiple sclerosis, SPMS—secondary progressive multiple sclerosis, PPMS—primary progressive multiple sclerosis, BNMS—benign multiple sclerosis, ON—optic neuritis, NON—non-optic neuritis. R—correlation coefficient calculated for CIS and MS patients adjusted for disease duration; r—Pearson correlation coefficient calculated for individual MS type controlled for the patients’ age.

## Data Availability

The datasets used and analyzed during this study are available from the corresponding author on reasonable request.

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
