# Peer review of "Spectral-Domain Optical Coherence Tomography Assessment in Treatment-Naïve Patients with Clinically Isolated Syndrome and Different Multiple Sclerosis Types: Findings and Relationship with the Disability Status"

_jcm, 2021, doi:10.3390/jcm10132892_

Round 1

Reviewer 1 Report

The authors have addressed my points satisfactorily. They should add the following sentence to the description of the statistical analyses: 'Both eyes of the patients and controls were analysed.' The revised text now also says how they accounted for this fact in the analyses (with an intra-subject correlation term), which is good.

The authors should add this sentence, but the Editor or production staff can verify this - I do not need to review the manuscript again.

Author Response

We would like to thank reviewer for comments that helped us to improve the manuscript. Below please find the point-by- point response to reviewer’s comments.

Reviewer 1:

The authors have addressed my points satisfactorily. They should add the following sentence to the description of the statistical analyses: 'Both eyes of the patients and controls were analysed.' The revised text now also says how they accounted for this fact in the analyses (with an intra-subject correlation term), which is good.

The authors should add this sentence, but the Editor or production staff can verify this - I do not need to review the manuscript again.

Response: We thank the reviewer for valuable  comment. We have added the sentence indicated by the reviewer to the section ,,Statistical Analysis” (lines 148-149).

Reviewer 2 Report

This manuscript describes the peripapillary retinal nerve fibre layer (pRNFL) thickness and total macular volume (TMV) using OCT in various phenotypes of MS (CIS, RRMS, SPMS, PPMS, BNMS) as well as healthy controls. All patients were treatment naïve. Eyes were divided into those with and without a history of optic neuritis. All MS eyes showed reduced TMV and pRNFL thickness compared to controls, with BNMS and SPMS being lowest. In CIS, only eyes with ON had decreased TMV and pRNFL thickness. TMV and pRNFL in non-ON eyes correlated best with MS disability.

I think one of the key features in this study is that participants are all treatment naïve and therefore we’re looking at the natural history of MS in this group. These types of studies are becoming increasingly rare since there are more and more treatment options open to patients. That being said, there is the limitation that this will select only a certain type of patient who chose not to take treatment (as mentioned in the limitations).

Specific comments:

  1. Is there an age dependence to pRNFL thickness and TMV? This does not seem to be mentioned anywhere. A quick search seems to indicate that they decrease with age. I therefore wonder if some of the larger decrease in pRNFL and TMV in BNMS might be due to their older age.
  2. For the CIS population, there are factors that can predict risk of conversion to MS (e.g. presence of many lesions on MRI). Did you look at any of these factors? How likely is it that your CIS patients will become MS?
  3. Table 1, can you please include the age range for HCs.
  4. Table 1, is the median EDSS score for SPMS correct? I find it surprising that with a range of values between 4-6.5 that the median would be 4.3.
  5. Figure 2 and 3, I’m unsure why the authors decided to only mark the significant differences between non-ON eyes. Although it will make the figure busier, I think it would be good to mark all the significant differences in the figures. (I know that all the differences are listed in the text as well as some in Table 2 but I think it’s more intuitive to see them on the figure.)
  6. For the correlations with disease duration (section 3.4), I wonder if including a regression over all the participants together would be helpful since MS is sometimes thought of as a continuum of disease. By dividing the groups, you are limiting both BNMS and CIS to be significant because of their smaller dynamic range in disease duration.
  7. In section 3.4, are the correlations with disease duration using both eyes? Is there a difference between ON and non-ON eyes? I have the same question for the pRNFL vs TMV correlations.
  8. I think plots demonstrating the correlations listed in Table 3 would be useful. Maybe 2 plots, one with disease duration separated into the groups and one with EDSS separated into ON/non-ON eyes.
  9. Does a history of ON in one eye affect the OCT values in the other eye? It seems that you’re treating each eye as independent of each other and I wonder if this is valid. Also, is it fair to throw all participants together when some have a history of ON and others don’t? A much larger decrease in pRNFL and TMV will be found in people with ON history so they might skew the data. (For example, if all the people with ON history had longer disease duration, then a correlation might be found from this pooling of data.) The fact that each subgroup had a different percentage of ON eyes might also affect differences between subgroups.

Author Response

We would like to thank reviewer for comments that helped us to improve the manuscript. Below please find the point-by- point response to reviewer’s comments.

Reviewer 2:

This manuscript describes the peripapillary retinal nerve fibre layer (pRNFL) thickness and total macular volume (TMV) using OCT in various phenotypes of MS (CIS, RRMS, SPMS, PPMS, BNMS) as well as healthy controls. All patients were treatment naïve. Eyes were divided into those with and without a history of optic neuritis. All MS eyes showed reduced TMV and pRNFL thickness compared to controls, with BNMS and SPMS being lowest. In CIS, only eyes with ON had decreased TMV and pRNFL thickness. TMV and pRNFL in non-ON eyes correlated best with MS disability.

I think one of the key features in this study is that participants are all treatment naïve and therefore we’re looking at the natural history of MS in this group. These types of studies are becoming increasingly rare since there are more and more treatment options open to patients. That being said, there is the limitation that this will select only a certain type of patient who chose not to take treatment (as mentioned in the limitations).

Specific comments:

  1. Is there an age dependence to pRNFL thickness and TMV? This does not seem to be mentioned anywhere. A quick search seems to indicate that they decrease with age. I therefore wonder if some of the larger decrease in pRNFL and TMV in BNMS might be due to their older age.

Response: Correlation of RNFL thickness as well as TMV with age was calculated initially for HCs. However, the obtained values were not statistically significant (r=-0.18; p=0.0464 and r=-0.20; p=0.0293, respectively) according to the adopted cut-off point of p<0.01. Therefore, to better visualize the impact of CIS as well as different MS types on the above SD-OCT parameters, the correlations were performed using the disease duration instead of the patients' age. This information was added to the manuscript text (lines 249-252).

2. For the CIS population, there are factors that can predict risk of conversion to MS (e.g. presence of many lesions on MRI). Did you look at any of these factors? How likely is it that your CIS patients will become MS?

Response: We thank the reviewer for this important remark. In the presented study, we did not analyze the factors that can predict the risk of conversion from CIS to MS. This aspect exceeded the objectives of the study and would require taking into account the nature of the first symptoms, the MRI scans and the follow-up period. In the CIS group, we focused on the incidence of optic neuritis, which resulted from the study aims. The diagnosis of CIS was each time established according to the revised 2010 McDonald criteria as described in the ,,Materials and Methods” section (line 103-105). Moreover, the lack of evaluation of the activity of demyelinating lesions on MRI was included in the limitation of the study (lines 421-423).

3. Table 1, can you please include the age range for HCs.

Response: We thank the reviewer for this comment. We have included the age range for HCs (18-69 years) in Table 1.

4. Table 1, is the median EDSS score for SPMS correct? I find it surprising that with a range of values between 4-6.5 that the median would be 4.3.

Response: We thank the reviewer for this valuable remark. The median EDSS score in SPMS group was 4.3 but the correct range of EDSS score was 3-6.5 (this error has been corrected in the Table 1).

5. Figure 2 and 3, I’m unsure why the authors decided to only mark the significant differences between non-ON eyes. Although it will make the figure busier, I think it would be good to mark all the significant differences in the figures. (I know that all the differences are listed in the text as well as some in Table 2 but I think it’s more intuitive to see them on the figure.)

Response: We thank the reviewer for valuable  comment. The differences in TMV and RNFL values between ON eyes of patients with CIS and various MS types did not reach the statistical significance adopted for the study (p<0.01)(lines 193-196 and 226-228, respectively). For this reason, in Figures 1 and 2 only the differences between NON eyes were marked. Due to the immediate proximity of the ON and NON bars for individual MS types, the figure does not show significant differences between ON and NON eyes within individual variants.

6. For the correlations with disease duration (section 3.4), I wonder if including a regression over all the participants together would be helpful since MS is sometimes thought of as a continuum of disease. By dividing the groups, you are limiting both BNMS and CIS to be significant because of their smaller dynamic range in disease duration.

Response: We thank the reviewer for this valuable comment. A characteristic feature of MS is the high hetereogeneity of its individual variants, both in terms of the disease duration, the nature of the symptoms and the disability progression, which was reflected in the investigated groups. In the presented study, CIS was not considered as MS variant. Therefore, both the title and the objectives of our study separate the CIS group from different MS variants. For this reason, no regression model was performed for all patients studied. Another aspect is the uncertainty that all CIS patients will convert to MS, which has already been raised by the reviewer above.

7. In section 3.4, are the correlations with disease duration using both eyes? Is there a difference between ON and non-ON eyes? I have the same question for the pRNFL vs TMV correlations.

Response: We thank the reviewer for this remark. The correlations with disease duration used both eyes of the patients with an intra-subject correlation term. Therefore, we have added the sentences: „Both eyes of the patients and controls were analysed. When verifying the statistical significance of differences between selected study groups, mixed-effects generalized linear models with intra-subject correlation term were fitted. Correlation coefficients were estimated by using a mixed-effects linear regression model with intra-subject correlation term.” to the section ,,Statistical Analysis” (lines 157-161). Taking into account the fact that in the CIS group there were only 3 ON eyes (10%) and in the PPMS group only 1 ON eye (4.55%), no separate correlations of disease duration with SD-OCT parameters were made for ON and NON eyes in individual groups. Therefore, the correlations of EDSS score with TMV and pRNFL values were performed for ON i NON eyes without distinguishing disease types. Moreover, in both ON and NON eyes, a similar correlation strength was found between pRNFL thickness and disease duration (r=-0.36; p<0.0001 and r=-0.34; p = 0.006, respectively). Furthermore,  significant correlations of TMV with disease duration were found for both ON and NON eyes (r=-0.38; p<0.0001 and r=-0.34; p=0.006, respectively). Significant correlations between pRNFL thickness and TMV were found both for ON eyes (r=0.60, p<0.0001) and NON eyes (r=0.62, p<0.0001). These data were added to the text of the manuscript (lines 255-258 and 266-268). All the correlations mentioned above show statistically significant differences according to the grouping variable of ON (everywhere at the level of p<0.001). Despite the similar values ​​of the correlation coefficients, we observe a statistically significant difference. The reliability of the calculations is increased by the size of the study group and the measurements of high accuracy and quality.

8. I think plots demonstrating the correlations listed in Table 3 would be useful. Maybe 2 plots, one with disease duration separated into the groups and one with EDSS separated into ON/non-ON eyes.

Response: We thank the reviewer for this remark. We have added two Figures to the manuscript that show significant relationships between TMV and disease duration (obtained for BNMS and PPMS patients - Figure 4) as well as significant relationships between TMV and EDSS score for ON and NON eyes (Figure 5).

9. Does a history of ON in one eye affect the OCT values in the other eye? It seems that you’re treating each eye as independent of each other and I wonder if this is valid. Also, is it fair to throw all participants together when some have a history of ON and others don’t? A much larger decrease in pRNFL and TMV will be found in people with ON history so they might skew the data. (For example, if all the people with ON history had longer disease duration, then a correlation might be found from this pooling of data.) The fact that each subgroup had a different percentage of ON eyes might also affect differences between subgroups.

Response: We thank the reviewer for this valuable remark. Taking into account both  of the patient's eyes, the statistical methods were adjusted as mentioned above: ,,Both eyes of the patients and controls were analysed. When verifying the statistical significance of differences between selected study groups, mixed-effects generalized linear models with intra-subject correlation term were fitted. Correlation coefficients were estimated by using a mixed-effects linear regression model with intra-subject correlation term” (lines 157-161).  The mentioned MS heterogeneity is also reflected in the different percentage of patients with individual disease variants who experience ON. For this reason, an episode of ON in patients with PPMS is extremely rare, as in our group. In order to present the diversity of the natural MS course of the disease, we were satisfied that we could keep the proportions similar to those found in everyday medical practice. In this context, selecting groups with a similar percentage of ON eyes would not reflect the natural course of the disease and would be of limited clinical value.
